# Effect of Tank Size on Zebrafish Behavior and Physiology

**DOI:** 10.3390/ani10122353

**Published:** 2020-12-09

**Authors:** Abudusaimaiti · Maierdiyali, Lin Wang, Yunchao Luo, Zhongqiu Li

**Affiliations:** 1Lab of Animal Behavior and Conservation, School of Life Sciences, Nanjing University, Nanjing 210023, China; ma20ab@163.com (A.M.); wanglin_nwsuaf@outlook.com (L.W.); luoyunchao@outlook.com (Y.L.); 2Center for Nature and Society, School of Life Sciences, Peking University, Beijing 100871, China

**Keywords:** zebrafish, tank size, behavior change, sex difference, physiology

## Abstract

**Simple Summary:**

Living space is an important aspect of animal welfare. Understanding the effects of welfare on experimental animals would help in drawing a precise conclusion. In this work, zebrafish in different tank sizes were studied through behavioral and physiology tests. Results showed that changes in the tank size affected zebrafish behavior; those that lived in small tanks behaved less boldly, had poor stamina, and spent much time on movement. Therefore, researchers should focus on zebrafish’s living space to generate valid data from laboratory studies.

**Abstract:**

Environmental conditions strongly affect experimental animals. As a model organism, zebrafish has become important in life science studies. However, the potential effect of living environment on their behavior and physiology is often overlooked. This work aimed to determine whether tank size affects zebrafish behavior and physiology. Tests on shelter leaving, shelter seeking, shoaling, stamina, and pepsin and cortisol levels were conducted. Results showed that zebrafish behavior is easily affected by changes on the tank size. Fish that lived in small tanks behaved less boldly, had poor stamina, and spent much time on movement. Sex differences in behavior were only evident in the shelter seeking tests. Tank size had no effect on pepsin and cortisol, but cortisol concentrations in males were lower than those in females. This study suggests that zebrafish behavior is easily influenced by their living environment, and future related studies should consider their living space.

## 1. Introduction

Experimental animals play an important role in science [1]. Enrichment can be beneficial for experimental animals held in captivity [2,3], but not in all cases [4,5]. Poor animal welfare may affect scientific validity and repeatability of experiments [6]. The size of living space also influences animal welfare [7]. Small enclosures trigger repetitive and invariant behaviors in some experimental animals. [8,9,10].

Captive animal welfare may improve their health and natural behavior expression [6]. Recent investigations on fish living space suggested that growth and feed intake increase with the tank size [11], and big enclosures make fish swim faster than usual [12]. Koi carp *(Cyprinus carpio)* in big living space grows better and has higher survival rate than those in small spaces [13].

Zebrafish has become an important model organism because of its strong reproduction ability, ease of feeding, small size, and transparent embryo [14,15]. This animal plays a key role in answering ethological [16], physiological [17], biomedical [18], and ecological [19] research questions. In general, a quarantine system with the following parameters is recommended for fish rearing: adjusted rearing practices, maintained temperature within 24–29 °C range, and use of a standardized static dark–light cycle [20]. Research on zebrafish husbandry revealed that changes in the water or tank may induce their stress [21]. Providing enrichment to housed zebrafish could improve their welfare and decrease their anxiety [22].

Fish used in the laboratory are commonly housed in barren tanks [23]; however, the effects caused by using different tank sizes are ignored, thus contributing to biased and conflicting outcomes [24]. Negative welfare in fish could be assessed by behavior, include changes in shoaling, aggressiveness, or the presence of stereotypic behavior [25,26]. The average activity of mosquitofish *(Gambusia holbrooki)* increases with tank size [27], and zebrafish *(Danio rerio)* show high locomotor activity when housed in large tanks [28].

Tests were conducted to determine whether the size of living space could influence zebrafish behavior and psychology. Fish boldness can be reflected by the length of stay in an open area [29]. Hence, shelter leaving and shelter seeking tests were performed to observe how long it would take for the fish to explore or leave the open area. Shoaling is one of the few naturalistic behaviors documented in wild zebrafish [30] and was evaluated in this work by conducting a shoaling test. A small tank size could restrict the free swimming of fish; therefore, a stamina test was designed to determine whether the fish’ swimming capability is affected by a small space. One of commonly used laboratory measures for animal welfare is plasma cortisol level [20] and the amount of free space for an individual is positively correlated with energy digestibility [31], so cortisol levels and pepsin activity were measured. Every fish was videotaped during feeding to identify the correlation between these test results and the fish character. An active fish might behave well during the test.

This study aims to investigate the influence of tank size on the behavior and physiology of zebrafish. Most fish feel stress when kept in unnatural conditions [32], and this stress leads to behavioral and physiological changes [33]. Our hypothesis is that the zebrafish living in small tank would: (i) behave less boldly, (ii) have low swimming capability, (iii) show high tendency to associate with a shoal, and (iv) have high cortisol level and low pepsin activity.

## 2. Materials and Methods

### 2.1. Animals and Treatments

Zebrafish of approximately 3 months old with a standard length of 3 ± 0.5 cm and a body weight of 0.4 ± 0.1 g were obtained from a commercial aquaculture and housed in tanks with water temperature of 25 °C inside an unmanned laboratory. The pH of water in every tank was monitored at the range of 7.5–8.2 three times during the feeding period. The fish were fed every other day, and 50% of the water was changed after 5 h to clean out excrement. Prior to the experiment, all zebrafish were kept in a big tank for 1-week acclimatization. Sick fish were removed from the tank. Experimental parameters are shown in Table 1.

Ten male and ten female zebrafish were randomly assigned to a control group for the physiology test, which was conducted to evaluate whether the changes in tank induce stress in zebrafish. The remaining fish were randomly categorized into four experimental groups, namely, groups A (mini tank) and B (small tank) with 14 tanks each and groups C (medium tank) and D (large tank) with 12 tanks each. Weight, shelter leaving, shelter seeking, and shoaling tests were performed before the fish were moved to their tanks. The animals remained in their respective tanks for 3 weeks. Every tank has one male and one female fish to ensure that every fish is identified in the statistical analysis; placing a single fish in the tank may cause isolation stress [34].

In accordance with Brydges and Braithwaite [35] experiment, the zebrafish in four groups were fed for 3 weeks. Behavior and physiology tests were conducted on each zebrafish after their weight was measured. This study was conducted in accordance with the Ethics Review Committee of Nanjing University (No. 2009-116). All applicable international, national, and institutional guidelines for the care and use of animals were followed. This article does not involve any studies with human participants.

### 2.2. Activity Pattern

Each tank was continuously videotaped for 30 min during feeding, and all fish were videotaped at noon to avoid the potential effect of diurnal rhythm. The camera was positioned directly above the tank to ensure that the movement of each fish could be seen. Time spent frozen was denoted as stationary, and the states of swimming, turning, and swinging were denoted as movement.

### 2.3. Behavior Test

#### 2.3.1. Shelter Leaving Test

Shelter leaving test was adopted from the study of Ullah, et al. [36]. Boldness was measured as the latency to leave the bucket. The water depth of test tank was 10 cm. A stone and a blocked net were placed in the shelter bucket to prevent floating and prohibit the fish from swimming out, respectively (Figure 1).

First, the individual was placed in a bucket and then in the test tank for 2 min to adapt to the new item. Timer was started when the blocked net was removed and then stopped when the fish was completely out of the bucket. If the zebrafish did not leave the bucket within 5 min, then the fish was noted as failing to complete the timeline.

#### 2.3.2. Shelter Seeking Test

Shelter seeking was defined as the latency to reach the plant. The water depth of test tank was 10 cm. The test tank was divided into two parts by using an opaque plate: one side contains the aquatic plants, and the other side was empty (Figure 1).

The individual was placed on the empty side for 2 min of acclimatization. Timer was started when the opaque plate was removed and then stopped when the subject swam into or touched the plant. If the zebrafish did not reach the plant within 5 min, then the fish was noted as failing to complete the timeline.

#### 2.3.3. Shoaling Test

The shoaling test was adopted from the study of Bergendahl, et al. [37]. The water depth of test tank was 7 cm, and transparent and opaque plates were placed close to each other in the middle of the test tank. The experimental fish was placed on the side of opaque plate, and another 10 zebrafish were placed on the side of transparent plate. The tested fish were not familiar with the fish from the shoal. The funnel only allowed the experimental individual to pass through the hole from one side to the other but prevented the fish school from entering the other side (Figure 1).

The individual was placed in the test tank for 2 min to adapt to environment, the opaque plate was then removed, and the timer was started. The experimental individual was encouraged to pass through hole by a school of fish in the other side. When the individual successfully passed through hole to the other side, the timer was stopped. The timeline was set as 10 min because the fish needed some time to find the hole.

#### 2.3.4. Stamina Test

A water pump was used to simulate the flow in an annular flume. The flow rate ranged from 35 mL/s (gear 1) to 135 mL/s (gear 100) (Figure 1).

The individual was placed in the annular water flume, and the flow rate was first set at 35 mL/s to allow the fish to swim upstream in the water flume. The rate was then slowly increased until the fish cannot persist and was washed away for at least two cycles. The flow rate was recorded as the stamina value.

### 2.4. Physiology Test

Cortisol level increases with physiological and environmental stress [38], and space reduction is a stressful influencing factor in animals [39]. Here, cortisol concentrations were measured to detect stress in fish, and pepsin activity was analyzed to determine whether stress is correlated with energy digestibility [31]. Given that zebrafish are small species, a large amount of sample must be obtained to quantify their cortisol levels [40,41]. In addition, measuring the cortisol level in the water may increase the fish’ stress level due to environmental changes and isolation stress [34].

The experimental fish were rapidly immersed in ice slurry to induce cold shock [42] and then taken out and cut in their abdomen. Viscera were separately dissected in tubes, kept in an ice box, and sent to Nanjing Jiancheng Institute of Biological Engineering within 5 h. The samples were cryopreserved immediately with liquid nitrogen.

Prior to the cortisol test, the samples were thawed at 2–8 °C. After cutting, the samples were weighed and added with PBS with the ratio of 1 g of tissue to 9 mL of PBS. Complete homogenization (SK-1, Guohua, China) was performed, followed by centrifugation (80-2, Shanghai, China) for approximately 20 min (2000–3000 rpm). The supernatant was collected, and cortisol concentrations were measured using a commercial ELISA kit (201905, Jiancheng, China) following the manufacturer’s instructions.

Prior to the pepsin activity test, the tissue was weighed precisely and then added with physiological saline with the ratio of 1 g of tissue to 9 mL of saline. The mixture was homogenized (SK-1, Guohua, China) to 10% homogenate and centrifuged (80-2, Shanghai, China) at 2500 rpm for 10 min. Pepsin activity was measured by using a Pepsin Assay Kit (A045-2, Jiancheng, China) following the manufacturer’s instructions.

### 2.5. Statistical Analysis

Each fish was only tested once to avoid familiarization. In the tests of shelter leaving, shelter seeking, and shoaling, data from the fish that did not complete the tests were considered as maximized time. The finish time (sec) of each individual was log10 transformed to avoid the influence of extreme values.

Correlations among weight, activity pattern, and each test result were examined for each group. Paired sample T test was conducted for the results of weight, shelter leaving, shelter seeking, and shoaling tests. Two-way ANOVA and chi-square test were performed for all second-round behavior, physiology, and activity pattern assessments with group and sex as factors. IBM SPSS Statistics 22.0 (SPSS Inc., Chicago, IL, USA) was used for data processing. Cox regression and mixed linear models (group and sex as independent fixed factors) were adapted to evaluate the relationship between test result and sex in R v3.5.1 with survival and nlme packages. Significant differences were reported at *p* < 0.05.

## 3. Results

### 3.1. Effect of Tank Size on Behavior

No correlation was found between weight and each behavior test. The results of paired sample T test revealed that after being raised separately in their respective tanks, individuals spent less time in shelter seeking test in mini tank (t_13_ = 3.034, *p* = 0.010), shelter leaving test in small (t_15_ = 2.620, *p* = 0.019) and big tanks (t_14_ = 4.647, *p* = 0.001), and shoaling test in small tank (t_15_ = 3.864, *p* = 0.002). All other groups also spent less time in these tests, but the difference was not significant (Table A1).

Two-way ANOVA results showed significant differences on shelter leaving test (F_3,53_ = 4.697, *p* = 0.006, Figure 2a), stamina test (F_3,53_ = 3.968, *p* = 0.013, Figure 2d), and activity pattern (F_3,53_ = 8.964, *p* < 0.001, Figure 2e) between each group. When the housing tank was large, the time to complete the shelter leaving test and the time spent for movement were short, and the upstream velocity was high. No difference was found in the results of shelter seeking test (F_3, 53_ = 0.684, *p* = 0.566, Figure 2b) and shoaling test (F_3, 53_ = 1.029, *p* = 0.387, Figure 2c) between each group.

The Cox regression model revealed that the female fish from the mini tank showed significant differences in activity pattern compared with all groups (small tank: z_57_ = 3.703, *p* < 0.001; medium tank: z_57_ = −3.892, *p* < 0.001; big tank: z_57_ = 2.781, *p* = 0.005; Table A2).

Chi-square analysis results for between the fish that finish and did not finish the tests revealed that the passing rate differed significantly between groups during shoaling test (χ^2^ = 8.604, df = 3, *p* = 0.035). The small tank and medium tank groups had higher passing rate than the large tank group. However, no difference was observed in the passing rate during shelter leaving (χ^2^ = 3.413, df = 3, *p* = 0.332) and shelter seeking tests (χ^2^ = 5.919, df = 3, *p* = 0.116).

### 3.2. Effect of Tank Size on Physiology Test

No correlation was found between the physiology and behavior of zebrafish. However, weight (r = −0.347, *p* = 0.006) was significantly correlated with pepsin activity but not with cortisol concentrations. No difference in pepsin activity (F_4, 81_ = 0.701, *p* = 0.593, Figure 3g) and cortisol concentrations (F_4, 81_ = 1.583, *p* = 0.187, Figure 3f) was observed between the experimental and control groups.

### 3.3. Influence of Sex

Chi-square analysis results showed that the passing rate of females was significantly higher than that of males in the shelter seeking test (χ^2^ = 5.762, df = 1, *p* = 0.016) but not in the shelter leaving (χ^2^ = 1.369, df = 1, *p* = 0.242) and shoaling tests (χ^2^ = 0.056, df = 1, *p* = 0.813).

Two-way ANOVA results showed that male fish spent more time on shelter leaving (F_1,53_ = 4.492, *p* = 0.039, Figure 2a) and shelter seeking tests (F_1,53_ = 5.006, *p* = 0.029, Figure 2b) than female fish. Moreover, the cortisol concentrations in males were significantly lower than those in females (F_1,81_ = 5.202, *p* = 0.025, Figure 3f), but their pepsin activity did not differ (F_1,81_ = 0.074, *p* = 0.786, Figure 3g).

Mixed linear model revealed that the male fish had significantly lower cortisol concentration (t_60_ = −2.091, *p* = 0.041, Table A3) than the females.

## 4. Discussion

### 4.1. Effect of Tank Size on Zebrafish Behavior

Results showed that shelter leaving, shelter seeking, and shoaling behaviors are all affected after feeding in different groups. The zebrafish living in small tanks behaved less boldly and had poor stamina. Although individual body size strongly affects zebrafish behavior [43], no correlation was found between these factors.

Paired sample T test revealed that changes in the tank led to different shelter leaving, shelter seeking, and shoaling test results compared with those before grouping. This finding indicated that changes in tank size alters the zebrafish behavior. Thus, we suggest to avoid changing the tanks of zebrafish in behavior studies.

In this work, the zebrafish living in large tanks act bolder than the others. Although a large number of studies were conducted on fish behavior, the contribution of environment to their behavioral response remains unclear [12,28,44,45]. Fish boldness can be reflected by some types of behavior [46], one of which is the length of stay in an open area [29]. In this experiment, the fish housed in a small tank spent a long time in the shelter leaving test but a short time in the shoaling test to avoid the open area. Therefore, a small tank size decreases the boldness of zebrafish.

According to the video recording, the zebrafish living in a small tank spent more time (higher than 90%) for movement than those living in a large tank. We hypothesized that a small space might trigger the stereotypical behavior of zebrafish. This concept is similar to how a small enclosure affects the behavior of other experimental animals, such as primates [8], rodents [9], and birds [10].

In the stamina test, the maximum upstream speed of zebrafish was lower in the small tank group than in the large tank group. However, the video recording of zebrafish activity showed that those living in a small tank were highly active. This finding indicated that the results of stamina test are not related to exercise. Therefore, we speculated that the difference in living tank size could affect the physique of zebrafish. When the housing tank is large, the fish will have good stamina.

### 4.2. Effect of Tank Size on Zebrafish Physiology

No difference in physiology was found among the four groups. Given that space reduction is an stressful influencing factor in animals [39], the zebrafish in small tanks should have high cortisol levels. Wilkes, Owen, Readman, Sloman and Wilson [23] reported that the cortisol of zebrafish is affected by environmental changes only during the first day and then tends to be normal afterward. This finding explains the lack of differences in the measured cortisol levels after 3 weeks of changing tanks. Therefore, spatial differences may only cause temporary stress in zebrafish but does not affect their cortisol level in the long run. In addition, temporary stress state does not have a long-term influence on other physiological functions such as pepsin activity in zebrafish.

### 4.3. Sex Differences

In the shelter seeking test, males spent more time in open area than females; this behavior is a sign of boldness and is consistent with the result that males are bolder than females [47,48,49,50]. Boldness was measured by the length of stay in an open area [29]. Hence, spending less time in shelter leaving test or more time in shelter seeking test reflects boldness. Shelter leaving test aims to determine the willingness of the fish to explore an open area, and shelter seeking test tries to identify the inclination of the animal to stay in an open area. Therefore, these two tests can measure boldness at both ends of the spectrum.

Male zebrafish tend to associate with a shoal compared with females [43], but this study did not achieve the same result. The reason might be because males show no preference when given the choice of groups with mixed sex [30], which was the stimulus shoal in the test.

Zebrafish’ cortisol levels were higher in females than in males. Although never been reported in zebrafish, sex difference in cortisol level has been observed in other species [51,52]. Cortisol is a useful indicator of zebrafish’s stressor [41]. Therefore, zebrafish females are also more anxious than males in captive condition.

## 5. Conclusions

Changes in the tank size could affect zebrafish behavior, and those living in small tanks behaved less boldly and had poor stamina. Activity pattern suggested that small space may trigger the stereotypical behavior of zebrafish. However, the feeding tank size [53,54] is rarely mentioned in experimental studies on zebrafish. If the lack of a specific environmental condition affects fish, then this would be a confounding factor and might lead to erroneous experimental conclusions [6,55]. Our study showed that living size is an important factor affecting captured zebrafish for research. Therefore, we recommended that living conditions should be considered. Small spaces should be avoided, and the living space must be kept unchanged during experiments involving zebrafish to avoid deviations in comparison.

No difference in physiology was found among the groups housed in different tank sizes, but the cortisol concentrations in males were lower than those in females. Specific and multiple physiology test must be further performed to assay the effects of tank size on the physiology of fish. Additional data should also be collected to analyze sex differences in behavior.

## Figures and Tables

**Figure 1 animals-10-02353-f001:**
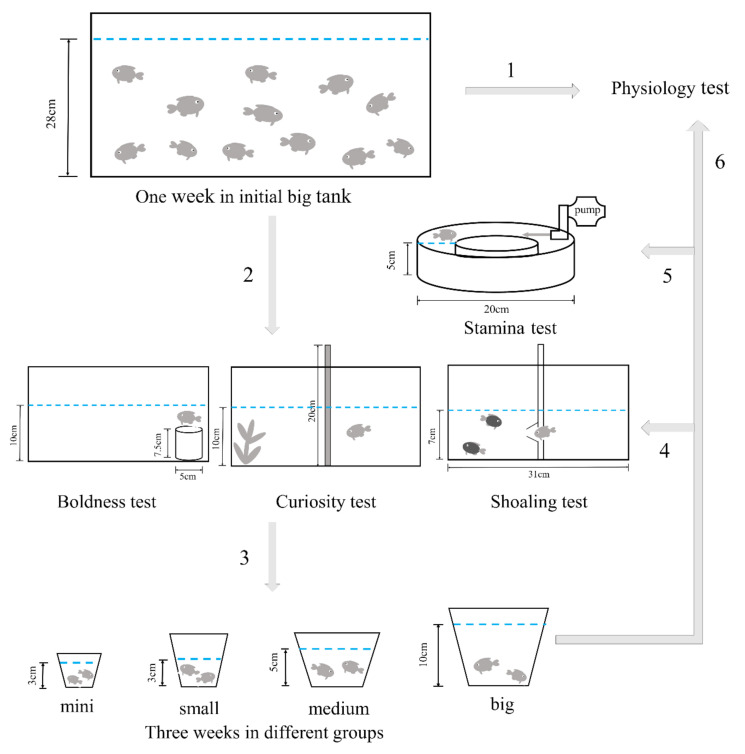
Experimental flow chart: 1. Physiology test for the control group. 2. Three behavior tests for the experimental fish. 3. Experimental fish were classified in four different groups. 3. Three behavior tests were performed again. 5. Stamina test for experimental fish. 6. Physiology test for experimental fish.

**Figure 2 animals-10-02353-f002:**
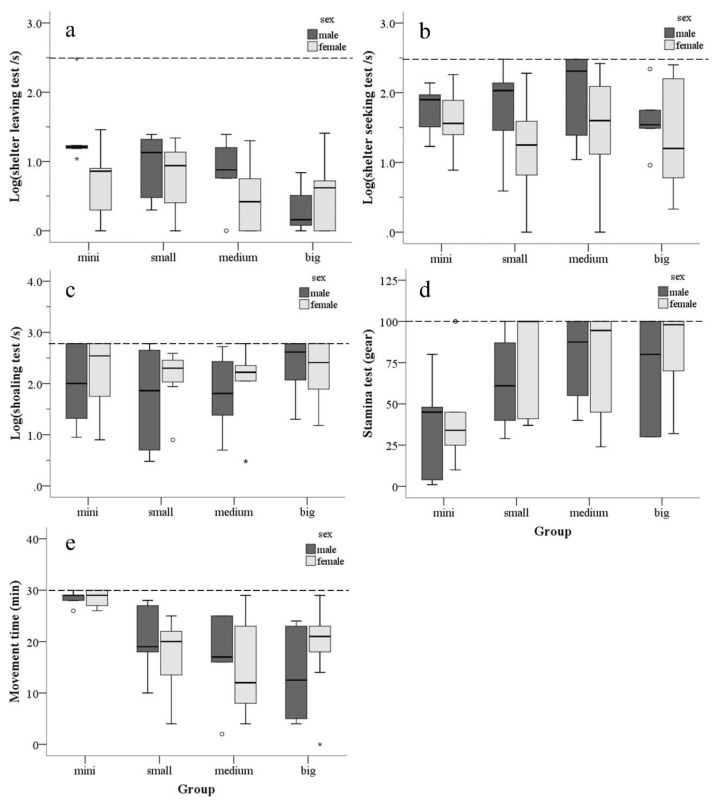
Test results for each group of males and females in each test. Abscissa: the size of different tank. Ordinate: (**a**) shelter leaving test time take logarithmic transformation (log_10_); (**b**) shelter seeking test time take logarithmic transformation (log_10_); (**c**) shoaling test time take logarithmic transformation (log_10_); (**d**) maximum gear for stamina test; and (**e**) activity pattern in half an hour. Dotted line: the maximum value. °: outlier points. *: extreme points.

**Figure 3 animals-10-02353-f003:**
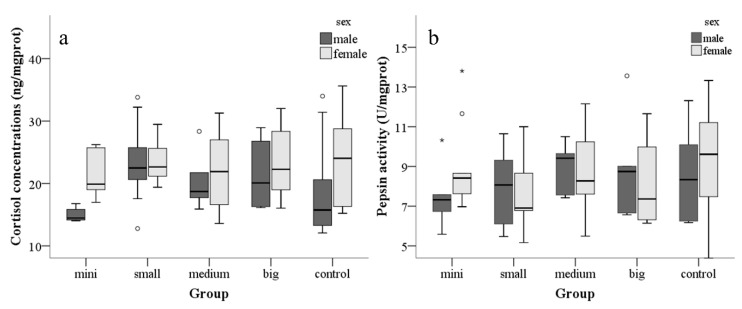
Average physiology test results for each group of males and females. Abscissa: mini tank, small tank, medium tank, large tank, control group. Ordinate: (**a**) cortisol concentrations, (**b**) pepsin activity. °: outlier points. *: extreme points.

**Table 1 animals-10-02353-t001:** Experimental parameters.

Items	Length/cm	Width/cm	Height/cm	Depth of Water/cm	Volume of Water/cm^3^
Mini tank A*	3.6	4.8	3.5	3	40
Small tank B *	5	6.5	7.5	3	80
Medium tank C *	8	11.8	7.5	5	400
Large tank D *	12.5	17.5	13.5	10	1500
Initial big tank	50	25	31	28	3500
Test tank	31	23	16		
Opaque plate		21	20		
Transparent plate **		21	20		
Water pump	75	50	78		
Annular flume	Inner diameter: 15	External diameter: 20	5	5	
Shelter		External diameter:5	7.5	7.5	

Note: * A, B, C, and D is cylinder-shaped tank. Hence, the length is equal to the bottom diameter, and the width indicates the up diameter. ** Transparent plate has a 3 cm diameter hole connected with a funnel. The edge of hole is 4 cm far from the bottom (similar to Figure 1 shoaling test).

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
