# Peer review of "Effect of Tank Size on Zebrafish Behavior and Physiology"

_animals, 2020, doi:10.3390/ani10122353_

Round 1

Reviewer 1 Report

The presented manuscript is interesting, concise and results seems to be reliable. However, some issues needs to be taken into consideration when revising the manuscript:

  1. In the introduction section some references to general recommendations for zebrafish husbandry should be added
  2. Line 75. Did the fish were quarantined prior experiment? What kind of ‘sickness’ were detected and could this influence the conducted experiment?
  3. In the Figure 1 caption A, B, C and D indicators needs to be explained
  4. L145 -146 this sentence needs to be rephrased
  5. L:149-150 It should be explained why cortisol and pepsin activity were measured because in the introduction section it is only briefly mentioned.
  6. Fig 2 and 3 do not include any statistical indicators. I am aware that the results were presented in logarytmic scale but it is hard to imagine which differences are significant if they are not presented in the graphs.
  7. L 217-218 this sentence needs to be rephrased.
  8. L219. This is obvious!
  9. L 222-224 I did not find any results concerning environmental enrichment in this study, so why mention it?
  10. L257-258. Is this sentence referring to human behavior? I hope not.
  11. L 256-261. highly unlikely explanation.
  12. I think that conclusions are a bit exaggerated. If there are no differences between the experimental groups or the difference are small it doesn't mean that the whole research is insignificant. In this case rather than follow some well-known trends it will be worth to look closer at the results. It is very beneficial for the reserchers that the tank do not influence strongly zebrafish behaviour and phisiology, and this fish is easily to adopt even to not optimal conditions which in rewiever’s opinion is verry nice conclusion. Of course it is always better to keep the fish in bigger tank, if possible. Moreover some more precise recommedndations should be included in this section.

Author Response

To Reviewer #1:

The presented manuscript is interesting, concise and results seems to be reliable. However, some issues needs to be taken into consideration when revising the manuscript:

In the introduction section some references to general recommendations for zebrafish husbandry should be added.

Reply: Thanks to your suggestion, we added “Generally, it is recommended to use a quarantine system for raising fish, adjust the rearing and maintenance temperatures within a 24–29°C range, and use a standardised static dark–light cycle.” and related references on line 44-45.

Line 75. Did the fish were quarantined prior experiment? What kind of ‘sickness’ were detected and could this influence the conducted experiment?

Reply: Yes, the fish were quarantined all time and we have added this information on line 74. The sickness here is that fish lied on the bottom of the tank and swum slowly, but we didn’t know for sure what kinds of illness it was.

In the Figure 1 caption A, B, C and D indicators needs to be explained

Reply: We have changed the A to mini, B to small, C to medium, D to big.

L145 -146 this sentence needs to be rephrased

Reply: Done. The sentence has been rephrased to “measuring the cortisol present in water may increase fish’s stress level due to the novelty of the environment and isolation stress.”

L:149-150  It should be explained why cortisol and pepsin activity were measured because in the introduction section it is only briefly mentioned.

Reply: We have added “Cortisol level would increase with increased physiological and environmental stress, and space-reduction is a stressful influencing factor in animals, so we measured cortisol concentrations to detect stress in fish and measured pepsin activity to see whether stress is correlated with energy digestibility” on line 146-148.

Fig 2 and 3 do not include any statistical indicators. I am aware that the results were presented in logarytmic scale but it is hard to imagine which differences are significant if they are not presented in the graphs.

Reply: We changed these figures to boxplots and did some improvement.

L 217-218 this sentence needs to be rephrased.

Reply: Done. The sentence has been rephrased to “This finding has indicated that changes in tanks induce changes in behavior, and we suggest to avoid changing the tanks of zebrafish in research.”

L219. This is obvious!

Reply: We think it is necessary to make readers understand our study is exploratory research.

L 222-224 I did not find any results concerning environmental enrichment in this study, so why mention it?

Reply: Thanks for your advice and we have deleted this part.

L257-258. Is this sentence referring to human behavior? I hope not.

Reply: We tried to explain it was the same with the result in human, but now we think it would be better not to mention it and changed it into “There was no sex differences reported in fish’s cortisol, but sex differences have been found in other species.”

L 256-261. highly unlikely explanation.

I think that conclusions are a bit exaggerated. If there are no differences between the experimental groups or the difference are small it doesn't mean that the whole research is insignificant. In this case rather than follow some well-known trends it will be worth to look closer at the results. It is very beneficial for the reserchers that the tank do not influence strongly zebrafish behaviour and phisiology, and this fish is easily to adopt even to not optimal conditions which in rewiever’s opinion is verry nice conclusion. Of course it is always better to keep the fish in bigger tank, if possible. Moreover some more precise recommendations should be included in this section.

Reply: Although the differences caused by tank size wasn’t found in every test, we still found several significant differences. We thought it is enough to tell researchers to report their raising environment especially the tank size when study zebrafish. Thank you for your suggestion,and our precise recommendation was included in conclusion.

Reviewer 2 Report

The paper by Maierdiyali et al investigates the effects of tank size on zebrafish behavioural and physiological traits. The authors find that there are some effects depending on tank size. Especially the largest and the smallest tanks had effects, as compared to other sizes (which one that had an effect depended on the test. The authors also report some sex-differences.

Overall, the study and the results is interesting from a zebrafish welfare perspective. The analyses need to incorporate the individuals that reached maximum time of the test – otherwise the data become biased! This is necessary for being acceptable for publication (see analysis suggestion below). Some additional minor issues are described below, which should be easy to address. The major focus of a revision should be to revise the writing. The structure of the text is characterized by quite short statements which makes the text “choppy” and affects the flow of reading. There are also some grammatical issues at some places. I’d like to state that the text is clearly readable, but I think sending it through a round of editing would increase the impression of professionalism. So, while I can read it without problems, I think the authors themselves might benefit from editing the language (in particular if any of the authors are students/post-docs, this may be beneficial in evaluations for future academic positions). It is no major critique about the writings skill, I’m mainly trying to give helpful advice. I have not suggested edits, as it would take too much time in relation to the short time-limit MDPI gives for reviews.

L31-33: I do not really understand why there is so much focus on environmental enrichment, when the paper is not about such modifications. Also, the authors mainly presents the studies showing positive effects from enrichment, while a substantial of other studies have shown no or negative effects – hence, the picture presented is biased. The paper should mainly focus on introducing previous studies on tank size, rather than structural enrichment. There are a few studies on this subject that could be used e.g. Polverino et al. 2016 (https://www.sciencedirect.com/science/article/pii/S0003347216000889), Espmark et al. 2017 (https://doi.org/10.1111/are.13121), etc…

L34: Here the authors state that enrichment is beneficial, but in many cases it is not. See e.g. Naslund & Johnsson 2016 (https://doi.org/10.1111/faf.12088), Johnsson et al. 2014 (https://doi.org/10.1111/jfb.12547).

L50: What is waving behaviour? I have done research on fish behaviour for over 10 years and never heard about this behaviour. Please explain in the text.

L54: Consider using ‘open’ or similar word instead of “blank”.

L70: “Fin-less body length” is really called “standard length” in fish research.

L74: Should be “clean out excrement”. Otherwise it sounds like the authors were polishing the excrements of the fish, which would be weird…

RESULTS (several places, but also described at L154): It seems like the fish reaching the maximum test time was removed from the data sets. This is not OK, because it creates a bias towards shorter performance time in each test. The fish that did not reach the goal during the short test time, would have made it eventually, with a longer time than the test time. Hence, these fish should be given the maximal time of the test (e.g. 5 or 10 min, depending on test). These data could then be analysed using time-to-event analyses (also called survival analyses; e.g. Cox regression – see e.g. the R-package ‘survival’ for the implementation of these types of tests). Survival analyses can handle censored data (i.e. when the maximal time has been reached, the individual is given the maximal time, but is also given a censor indication which is incorporated in the analysis) – this means that the analyses sees the individual as having e.g. “at least 5 min time to reach the plant, but unknown exactly how much longer”. For being acceptable for publication, this needs to be revised.

L118: How do the authors know that they measured curiosity, and not shelter-seeking? Fish typically seek shelter around structural objects. The fact that other studies may use this test for curiosity, does not mean that it is correct. It needs to be motivated or discussed.

L149-150: Please provide all the necessary details and materials used for the analyses of the physiological variables. As is, it is not enough. Much more detail is needed. A reader should be able to replicate the methods based on the description in the text.

L161: Weight is a continuous variable and, hence, cannot be a random factor. Specifying this as a random factor may give strange results.

Figure 2: Please consider presenting data as boxplots instead of bar-plots. Bar-plots are very bad at showing the variation in the data. Also, instead of labelling the groups A, B, C, D, why not just write the actual treatment as the label (mini, small, medium, large)? This would make the graph much easier to understand. Also, when a variable has been log10 transformed, write this in the label for the y-axis, e.g. log10(boldness score (s))

L185: present degrees of freedom for the t-test, as is done in other parts of the results.

L218: The author states that they find different results than before. Before what? This is unclear.

L222: It is not controversial that environment affects behaviour. The main problem is that behaviour researchers do not care…

L222-224: Different results from enrichment is not surprising. Different species reacts differently and different types of enrichments were used in the studies.

L248: The fact that males are known to be more bold in zebrafish makes me wonder whether the authors actually measured boldness in their test. Could it be that the test actually measured a different behaviour? Many researchers use the term *boldness* without knowing whether or not the measurement actually reflects boldness in a real-life situation. This is a major problem in the literature. Discussing whether the behaviour measured was actually boldness should be a part of the discussion here, to show that the authors actually understand the complexity in labelling simple test behaviours as e.g. “boldness”.

L258-261: This part is irrelevant. Comparing fish to mammals is not good practise (although many researchers do it to make their results seem more interesting, assuming that human/mammal relevance is important… another problem in the published animal behaviour literature). Different species have gone through different selection pressures, and may not show the same sex differences – hence, comparing with very different animal groups is largely problematic.

Author Response

The paper by Maierdiyali et al investigates the effects of tank size on zebrafish behavioural and physiological traits. The authors find that there are some effects depending on tank size. Especially the largest and the smallest tanks had effects, as compared to other sizes (which one that had an effect depended on the test. The authors also report some sex-differences.

Overall, the study and the results is interesting from a zebrafish welfare perspective. The analyses need to incorporate the individuals that reached maximum time of the test – otherwise the data become biased! This is necessary for being acceptable for publication (see analysis suggestion below). Some additional minor issues are described below, which should be easy to address. The major focus of a revision should be to revise the writing. The structure of the text is characterized by quite short statements which makes the text “choppy” and affects the flow of reading. There are also some grammatical issues at some places. I’d like to state that the text is clearly readable, but I think sending it through a round of editing would increase the impression of professionalism. So, while I can read it without problems, I think the authors themselves might benefit from editing the language (in particular if any of the authors are students/post-docs, this may be beneficial in evaluations for future academic positions). It is no major critique about the writings skill, I’m mainly trying to give helpful advice. I have not suggested edits, as it would take too much time in relation to the short time-limit MDPI gives for reviews.

Reply:Thank you very much for your interest and guidance in this study. We have carefully revised this manuscript according to your comments and suggestions. The specific modification is pointed out in the following text

L31-33: I do not really understand why there is so much focus on environmental enrichment, when the paper is not about such modifications. Also, the authors mainly presents the studies showing positive effects from enrichment, while a substantial of other studies have shown no or negative effects – hence, the picture presented is biased. The paper should mainly focus on introducing previous studies on tank size, rather than structural enrichment. There are a few studies on this subject that could be used e.g. Polverino et al. 2016 ( https://www.sciencedirect.com/science/article/pii/S0003347216000889), Espmark et al. 2017 ( https://doi.org/10.1111/are.13121), etc…

Reply: We have revised this part and focused on introducing previous studies on tank size. Thanks for your references.

L34: Here the authors state that enrichment is beneficial, but in many cases it is not. See e.g. Naslund & Johnsson 2016 ( https://doi.org/10.1111/faf.12088), Johnsson et al. 2014 ( https://doi.org/10.1111/jfb.12547).

Reply: Thanks for your suggestions and we have added explanation that enrichment is not beneficial in every case on line 32.

L50: What is waving behaviour? I have done research on fish behaviour for over 10 years and never heard about this behaviour. Please explain in the text.

Reply: We used waving to represent stereotypic behavior in the form of continuous circling, and we have deleted this word to avoid misunderstanding.

L54: Consider using ‘open’ or similar word instead of “blank”.

Reply: Done. We have changed “blank” into “open”.

L70: “Fin-less body length” is really called “standard length” in fish research.

Reply: Done. We have changed “Fin-less body length” into “standard length”.

L74: Should be “clean out excrement”. Otherwise it sounds like the authors were polishing the excrements of the fish, which would be weird…

Reply: Done. We have changed “clean excrement” into “clean out excrement”.

RESULTS (several places, but also described at L154): It seems like the fish reaching the maximum test time was removed from the data sets. This is not OK, because it creates a bias towards shorter performance time in each test. The fish that did not reach the goal during the short test time, would have made it eventually, with a longer time than the test time. Hence, these fish should be given the maximal time of the test (e.g. 5 or 10 min, depending on test). These data could then be analysed using time-to-event analyses (also called survival analyses; e.g. Cox regression – see e.g. the R-package ‘survival’ for the implementation of these types of tests). Survival analyses can handle censored data (i.e. when the maximal time has been reached, the individual is given the maximal time, but is also given a censor indication which is incorporated in the analysis) – this means that the analyses sees the individual as having e.g. “at least 5 min time to reach the plant, but unknown exactly how much longer”. For being acceptable for publication, this needs to be revised.

Reply: We have revised this according to your suggestions, and given the maximal time of test for fish that didn’t complete the test, and also used a Cox regression in R language (R-3.5.1) with survival package. Result was shown on line 205-208.

L118: How do the authors know that they measured curiosity, and not shelter-seeking? Fish typically seek shelter around structural objects. The fact that other studies may use this test for curiosity, does not mean that it is correct. It needs to be motivated or discussed.

Reply: We used the word “curiosity” just for another word to describe boldness. Actually, boldness test and curiosity test are both measure the length of stay in an open area. The related discussion on line 268-271. It might be more precise to named it “boldness 1” and “boldness 2”, but it would be easy to confused.

L149-150: Please provide all the necessary details and materials used for the analyses of the physiological variables. As is, it is not enough. Much more detail is needed. A reader should be able to replicate the methods based on the description in the text.

Reply: We have added all the details on line 155-172.

L161: Weight is a continuous variable and, hence, cannot be a random factor. Specifying this as a random factor may give strange results.

Reply: We found the weight is an irrelevant variable by previous analysis, so we deleted this factor in mixed line model. We shall be glad to receive your further suggestions about these models.

Figure 2: Please consider presenting data as boxplots instead of bar-plots. Bar-plots are very bad at showing the variation in the data. Also, instead of labelling the groups A, B, C, D, why not just write the actual treatment as the label (mini, small, medium, large)? This would make the graph much easier to understand. Also, when a variable has been log10 transformed, write this in the label for the y-axis, e.g. log10(boldness score (s))

Reply: Done. We have changed our presenting data as boxplots and taken all of suggestions.

L185: present degrees of freedom for the t-test, as is done in other parts of the results.

Reply: Done.

L218: The author states that they find different results than before. Before what? This is unclear.

Reply: We have made it clear that before means before dividing fish into different groups.

L222: It is not controversial that environment affects behaviour. The main problem is that behaviour researchers do not care…

Reply: Thank you for your advice. We have changed “controversial” into “unclear”.

L222-224: Different results from enrichment is not surprising. Different species reacts differently and different types of enrichments were used in the studies.

Reply: We have deleted this part of discussion about enrichment, considering our study is focus on tank size.

L248: The fact that males are known to be more bold in zebrafish makes me wonder whether the authors actually measured boldness in their test. Could it be that the test actually measured a different behaviour? Many researchers use the term *boldness* without knowing whether or not the measurement actually reflects boldness in a real-life situation. This is a major problem in the literature. Discussing whether the behaviour measured was actually boldness should be a part of the discussion here, to show that the authors actually understand the complexity in labelling simple test behaviours as e.g. “boldness”.

Reply: After giving the maximal time of test for fish that didn’t complete the test, statistical result showed males did show bolder than females. Discussion about how we measure the boldness in these tests on line 268-271.

L258-261: This part is irrelevant. Comparing fish to mammals is not good practise (although many researchers do it to make their results seem more interesting, assuming that human/mammal relevance is important… another problem in the published animal behaviour literature). Different species have gone through different selection pressures, and may not show the same sex differences – hence, comparing with very different animal groups is largely problematic.

Reply: We have deleted the part of comparing fish to mammals, and changed into “There was no sex differences reported in fish’s cortisol, but sex differences have been found in other species.”

Round 2

Reviewer 2 Report

L13: “changes in the tank fish affected zebrafish behaviour” is not a correct sentence

L18: “have” should be “has (“a model species” suggests zebrafish is in singular)

L33: I’d say “enrichment can be beneficial”, not “is usually beneficial”. There is no meta-analysis or standardized review on this subject, so the generality of this statement is not proven. The reviews that exist point to mixed results.

L35: “of experiments” rather than “in experiments”

L37: Good welfare may not always lead to natural behaviour. Natural behaviour includes aggression (sometimes lethal = not good welfare). Fear-responses, anti-predation and stress-responses are also natural behaviours, which are not seen in a good-welfare system.

L38: Replace “house” with “housed”

General comment: Please revise the English language. There are plenty of minor errors still in the text. I will make no further comments on this, because it takes too much time, and I see my task as a reviewer to concern the science, not the language – that’s a task for the authors and the copy editor.

L48: Why is there so much focus on enrichment, when this factor is not investigated by this study?

L63-64: This sentence seems incomplete. Given the relationship noted, what would be expected?

L120: Regarding usage of “curiosity” for a test measuring time to reach a refuge. Why is this test considered to measure curiosity and not simply shelter seeking? A fish that is frightened will be more quick to reach the refuge (the plant). The authors write in their response: “We used the word “curiosity” just for another word to describe boldness. Actually, boldness test and curiosity test are both measure the length of stay in an open area. The related discussion on line 268-271. It might be more precise to named it “boldness 1” and “boldness 2”, but it would be easy to confused.” The term cannot be used for mere convenience, there has to be a well-reasoned argument based on evidence before naming a behavioural trait.

L126/L137: What is actually meant by “failure” on these lines? It is not a failed test, just a fish reaching the ceiling time of the experiment. The only potential failure here is that the tests should have been longer – I mean no disrespect to the authors, short time frames for tests are common, but in future assays, the time probably needs to be increased to improve the conditions for reliable data analysis.

L156-173: These new lines read like an instruction manual, when they should read as what the authors did. If it was lacking information before, now there is excessive information. What I meant in my previous comment was not to describe every single step in the process, but to describe which Assay kits and what instruments and machinery was used in the assay. It is possible to just write that the manufacturers instructions were followed (if you write this, then I can replicate the analysis). In that case, only deviations from the standard protocol (if any) would be needed to be included. Please see other published papers analysing cortisol and pepsin for reference on appropriate level of information. The machines used are still not described, and details on manufacturer is needed for all materials (following general guidelines for writing scientific papers; see other papers for reference).

L181: Cox regression is really needed for all analyses including data where any fish reached maximum value, but ANOVAs are relatively robust, and I think it produces quite reliable results – qualitatively, but not necessarily quantitatively. For a really good analysis, choose Cox regression for all behavioural analyses concerning time-to-event data.

Fig 2: This looks much better now. Even better would be if the maximum value is denoted in the graph by a horizontal dotted line – this would make it clear where the ceiling value is located in the graph, and make interpretations easier.

L242: How does the authors know that it was the change in tank that led to the effects, and not tank size in itself? If a zebrafish is reared as a larvae in a very small tank, should it not be placed in a larger tank when it grows?

L252: Stereotypical behaviour was not really tested in the experiment. Higher activity may be indicative of stereotypical swimming, but it could also be an indicator of the fish looking to escape, or maybe general stress.

Author Response

Dear Editors and Reviewers:

Thank you for your letter and for the reviewers’ comments concerning our manuscript entitled “The effect of tank size on zebrafish behavior and physiology”. Those comments are all valuable and very helpful for revising and improving our paper, as well as the important guiding significance to our researches. We have studied comments carefully and have made correction which we hope meet with approval. Revised portion are marked in red in the paper. The main corrections in the paper and the response to the reviewer’s comments are as follows:

L13: “changes in the tank fish affected zebrafish behaviour” is not a correct sentence

Reply: The sentence has been rephrased to “changes in the tank size affected zebrafish behavior.”

L18: “have” should be “has (“a model species” suggests zebrafish is in singular)

Reply: Done.

L33: I’d say “enrichment can be beneficial”, not “is usually beneficial”. There is no meta-analysis or standardized review on this subject, so the generality of this statement is not proven. The reviews that exist point to mixed results.

Reply: The sentence has been rephrased to “Enrichment can be beneficial for experimental animals held in captivity.”

L35: “of experiments” rather than “in experiments”

Reply: Done.

L37: Good welfare may not always lead to natural behaviour. Natural behaviour includes aggression (sometimes lethal = not good welfare). Fear-responses, anti-predation and stress-responses are also natural behaviours, which are not seen in a good-welfare system.

Reply: The sentence has been rephrased to “Captive animal welfare improve their health and natural behavior expression.”

L38: Replace “house” with “housed”

Reply: Done.

General comment: Please revise the English language. There are plenty of minor errors still in the text. I will make no further comments on this, because it takes too much time, and I see my task as a reviewer to concern the science, not the language – that’s a task for the authors and the copy editor.

Reply: Thanks for your kindly advices. We will work on grammar to enable reviewers and future readers to more easily read.

L48: Why is there so much focus on enrichment, when this factor is not investigated by this study?

Reply: We have deleted enrichment part and added “The average activity of mosquitofish (Gambusia holbrooki) increases with tank size, and zebrafish (Danio rerio) show high locomotor activity when housed in large tanks”

L63-64: This sentence seems incomplete. Given the relationship noted, what would be expected?

Reply: The sentence has been rephrased to “Every fish was videotaped during feeding to identify the correlation between these test results and the fish character. An active fish might behave better well during then test.”

L120: Regarding usage of “curiosity” for a test measuring time to reach a refuge. Why is this test considered to measure curiosity and not simply shelter seeking? A fish that is frightened will be more quick to reach the refuge (the plant). The authors write in their response: “We used the word “curiosity” just for another word to describe boldness. Actually, boldness test and curiosity test are both measure the length of stay in an open area. The related discussion on line 268-271. It might be more precise to named it “boldness 1” and “boldness 2”, but it would be easy to confused.” The term cannot be used for mere convenience, there has to be a well-reasoned argument based on evidence before naming a behavioural trait.

Reply: Thank you very much for your suggestions. After discussing it with all authors, we have decided to change “boldness test” into “shelter leaving test”, and change “curiosity test” into “shelter seeking test.”

L126/L137: What is actually meant by “failure” on these lines? It is not a failed test, just a fish reaching the ceiling time of the experiment. The only potential failure here is that the tests should have been longer – I mean no disrespect to the authors, short time frames for tests are common, but in future assays, the time probably needs to be increased to improve the conditions for reliable data analysis.

Reply: We have changed “failure” into “failing to complete the timeline.” Thank you for your suggestions.

L156-173: These new lines read like an instruction manual, when they should read as what the authors did. If it was lacking information before, now there is excessive information. What I meant in my previous comment was not to describe every single step in the process, but to describe which Assay kits and what instruments and machinery was used in the assay. It is possible to just write that the manufacturers instructions were followed (if you write this, then I can replicate the analysis). In that case, only deviations from the standard protocol (if any) would be needed to be included. Please see other published papers analysing cortisol and pepsin for reference on appropriate level of information. The machines used are still not described, and details on manufacturer is needed for all materials (following general guidelines for writing scientific papers; see other papers for reference).

Reply: Thanks for your guidance. We have revised this part and described the machines.

L181: Cox regression is really needed for all analyses including data where any fish reached maximum value, but ANOVAs are relatively robust, and I think it produces quite reliable results – qualitatively, but not necessarily quantitatively. For a really good analysis, choose Cox regression for all behavioural analyses concerning time-to-event data.

Reply: Thank you for your advice. We have used a Cox regression to analysis behavior test, and used a mixed linear model to analysis physiology test. ANOVAs are used for every test.

Fig 2: This looks much better now. Even better would be if the maximum value is denoted in the graph by a horizontal dotted line – this would make it clear where the ceiling value is located in the graph, and make interpretations easier.

Reply: Done. A horizontal dotted line was added in every figure.

L242: How does the authors know that it was the change in tank that led to the effects, and not tank size in itself? If a zebrafish is reared as a larvae in a very small tank, should it not be placed in a larger tank when it grows?

Reply: Thanks to your advice, we have rephrased this sentence to “This finding indicated that changes in tank size alters the zebrafish behavior” According to our results, we know zebrafish behave differently in different tank size, so it would be more precise to say “changes in tank size.”

L252: Stereotypical behaviour was not really tested in the experiment. Higher activity may be indicative of stereotypical swimming, but it could also be an indicator of the fish looking to escape, or maybe general stress.

Reply: The results of cortisol test indicated that there was no stress difference in different tank, so we thought activity pattern shouldn’t show differences if it was general stress. The fact that zebrafish spent more time in movement might be an indicator of the fish looking to escape, and the fish lived in smaller tank moved more frequently, so we just guessed it might become stereotype and said “We hypothesized that a small space might trigger the stereotypical behavior of zebrafish.”

With respect to the language, we have asked an english polishment company to re-edit the whole text, please find attached file with tracks, hope it better now.

This manuscript is a resubmission of an earlier submission. The following is a list of the peer review reports and author responses from that submission.

Round 1

Reviewer 1 Report

A thorough rewriting of manuscript is required as numerous syntax errors were spotted. Apart from this serious drawbacks were also found. For example in Table 1 volumes of water are wrong. 

It is very difficult for the reader to follow and understand the experimental procedure as presented in materials & methods. A more comprehensive description of the methodology should be given.